# Plasma Proteins Associated with COVID-19 Severity in Puerto Rico

**DOI:** 10.3390/ijms25105426

**Published:** 2024-05-16

**Authors:** Lester J. Rosario-Rodríguez, Yadira M. Cantres-Rosario, Kelvin Carrasquillo-Carrión, Alexandra Rosa-Díaz, Ana E. Rodríguez-De Jesús, Verónica Rivera-Nieves, Eduardo L. Tosado-Rodríguez, Loyda B. Méndez, Abiel Roche-Lima, Jorge Bertrán, Loyda M. Meléndez

**Affiliations:** 1Department of Microbiology and Medical Zoology, University of Puerto Rico, Medical Sciences Campus, San Juan 00935, Puerto Rico; lester.rosario@upr.edu; 2Translational Proteomics Center, Research Capacity Core, Center for Collaborative Research in Health Disparities, University of Puerto Rico, Medical Sciences Campus, San Juan 00935, Puerto Rico; yadira.cantres@upr.edu (Y.M.C.-R.); ana.rodriguez48@upr.edu (A.E.R.-D.J.); 3Integrated Informatics, Research Capacity Core, Center for Collaborative Research in Health Disparities, University of Puerto Rico, Medical Sciences Campus, San Juan 00935, Puerto Rico; kelvin.carrasquillo@upr.edu (K.C.-C.); eduardo.tosado@upr.edu (E.L.T.-R.); abiel.roche@upr.edu (A.R.-L.); 4Interdisciplinary Studies, Natural Sciences, University of Puerto Rico, Río Piedras Campus, San Juan 00925, Puerto Rico; alexandra.rosa2@upr.edu (A.R.-D.); veronica.rivera38@upr.edu (V.R.-N.); 5Department of Science & Technology, Ana G. Mendez University, Carolina 00928, Puerto Rico; lbmendez@uagm.edu; 6Infectious Diseases, Auxilio Mutuo Hospital, San Juan 00919, Puerto Rico; jorge.bertran@upr.edu

**Keywords:** COVID-19, SARS-CoV-2, cadherin-13, H-cadherin, T-cadherin, TNF-α, proteomics, Puerto Rico

## Abstract

Viral strains, age, and host factors are associated with variable immune responses against SARS-CoV-2 and disease severity. Puerto Ricans have a genetic mixture of races: European, African, and Native American. We hypothesized that unique host proteins/pathways are associated with COVID-19 disease severity in Puerto Rico. Following IRB approval, a total of 95 unvaccinated men and women aged 21–71 years old were recruited in Puerto Rico from 2020–2021. Plasma samples were collected from COVID-19-positive subjects (*n* = 39) and COVID-19-negative individuals (*n* = 56) during acute disease. COVID-19-positive individuals were stratified based on symptomatology as follows: mild (*n* = 18), moderate (*n* = 13), and severe (*n* = 8). Quantitative proteomics was performed in plasma samples using tandem mass tag (TMT) labeling. Labeled peptides were subjected to LC/MS/MS and analyzed by Proteome Discoverer (version 2.5), Limma software (version 3.41.15), and Ingenuity Pathways Analysis (IPA, version 22.0.2). Cytokines were quantified using a human cytokine array. Proteomics analyses of severely affected COVID-19-positive individuals revealed 58 differentially expressed proteins. Cadherin-13, which participates in synaptogenesis, was downregulated in severe patients and validated by ELISA. Cytokine immunoassay showed that TNF-α levels decreased with disease severity. This study uncovers potential host predictors of COVID-19 severity and new avenues for treatment in Puerto Ricans.

## 1. Introduction

Severe acute respiratory syndrome virus-2 (SARS-CoV-2) causes coronavirus disease 2019 (COVID-19), ranging from asymptomatic to severe disease. As of 17 March 2024, there were over 774 million confirmed COVID-19 cases and over 7 million deaths worldwide [1]. The development of coronavirus disease depends on interactions of viral and host responses and ranges from asymptomatic to severe disease and death. Patients with severe disease present a cytokine storm in the blood that may include elevation of TNF-α, CCL2, CXCL10 (IP-10), MCP-1, MIP-1α, GCSF, IL-1β, IL-2, IL-6, IL-7, IL-8, IL-10, and IL-17 [2,3,4,5,6,7,8,9]. A recent study using machine learning/artificial intelligence approaches in plasma proteomics datasets of COVID-19 patients linked severe disease with B cell dysfunction, increased inflammation, activation of Toll-like receptors, and decreased activation of developmental and immune mechanisms such as SCF/c-Kit signaling [10]. The proteins identified by artificial intelligence with the highest predictive values for COVID-19 disease severity were the following: CRK-like proto-oncogene, adaptor protein (*CRKL*), interleukin 1 receptor-associated kinase 1 (*IRAK1*), NF-kappa-B essential modulator/inhibitor of nuclear factor kappa-B kinase subunit gamma (NEMO/*IKBKG*), axis inhibition protein 1 (*AXIN1*), serine/arginine-rich protein-specific kinase 2 (*SRPK2*), and the cytoplasmic histidine–TRNA ligase (*HARS1*) [10]. Increased growth differentiation factor 15 (GDF-15) levels were positively associated with COVID-19 severity in a relatively large proteomic study in Canada [11]. In a recent study in the United States, increased expression of mesothelin (*MSLN*) in severe patients was identified by proteomics [12]. This protein has not been reported in previous studies of COVID-19 severity. People over 55 years old with comorbidities were more likely to have a severe COVID-19 disease outcome [6,13,14,15]. In a recent study in Estonia, the two factors that mostly correlated with disease severity were age and male gender [16]. However, a Mendelian randomization study using epigenetic clocks and telomere length found that aging is not a risk factor for COVID-19 disease severity and susceptibility [17].

Population-specific immune responses to SARS-CoV-2 have been recently identified [16]. The factors associated with disease severity are heterogeneous, and therefore, more population-based studies are needed [18]. New findings indicate that unvaccinated patients are at increased risk of disease severity [19]. Today, the COVID-19 pandemic has not ended because of the emergence of many new variants that started with Alpha, Beta, Gamma, Delta, and now, with additional variants of Omicron [1]. Given the increased number of COVID-19 cases, it remains to be understood which are the unique factors that protect the different populations in the world against disease severity. Quantitative proteomics is a powerful method to uncover novel proteins associated with different conditions. For COVID-19 disease, initial studies of plasma proteomics were developed in 2020 in Korea by comparing severe patients to mild ones and controls, which revealed important proteins associated with disease severity [20]. Similar studies were performed in China [21] and the United States [22]. This last study stratified patients according to their IL-6 levels. Most plasma and serum proteomics studies performed worldwide are summarized in Appendix A.

The purpose of this study is to perform proteomics analysis to identify the host factors that are associated with COVID-19 disease severity in Puerto Ricans. We hypothesized that unique proteins/pathways are associated with COVID-19 disease severity in Puerto Ricans. In summary, through proteomics analyses, we found cadherin-13 downregulated in severe patients, and these results were validated by ELISA. Cytokine analyses showed decreased TNF-α in severe patients. In addition to vaccination, these proteins represent unique targets for COVID-19 prevention and treatment of our population.

## 2. Results

### 2.1. Demographics

A total of 95 unvaccinated participants (*n* = 95) living in Puerto Rico were recruited in the study. Both men (41.1%) and women (58.9%) were included in the study in the 21–71 y/o age range. Participants with acute COVID-19 (*n* = 39; 41.1%) were included and compared to COVID-19-negative controls (*n* = 56; 58.9%). The COVID-19 questionnaire that was administered to all participants included the clinical information collected regarding the history of COVID-19 disease with 20 signs and symptoms; eight of these were described in this cohort. Results revealed that 18/39 (46.2%) of the participants, including both genders, had mild COVID-19; 13/39 (33.3%) had moderate, and 8/39 (20.5%) severe COVID-19 (Table 1). The most common comorbid conditions in this cohort were high blood pressure, autoimmune diseases, diabetes, and cardiovascular diseases. Only 3.2% of the participants reported being HIV positive and were excluded from the TMT analyses. There was a significant proportion of patients with cardiovascular disease in the severe COVID-19-positive group compared to the rest of the groups (*p* = 0.0005).

### 2.2. Proteomic Profile of Puerto Rican COVID-19 Patients

TMT quantitative analyses were performed with 30 patients stratified by mild, moderate, and severe, as described (Figure 1A,B). A human cytokine array, including twenty-three analytes, was performed to identify cytokines, chemokines, and growth factors associated with COVID-19 severity (Figure 1C). Proteomics analyses revealed unchanged and differentially expressed proteins, as shown in volcano plots for Mild vs. COVID-19 negative (Figure 2A), Moderate vs. COVID-19 negative (Figure 2B), and Severe vs. COVID-19 negative (Figure 2C). In mild COVID-19 patients, there were 6 upregulated and 3 downregulated proteins when compared to COVID-19-negative controls (Figure 2D); in moderate COVID-19 patients, there were 3 upregulated and 8 downregulated proteins (Figure 2D); and in severe COVID-19 patients, there were 2 upregulated and 54 downregulated proteins (Figure 2D). In total, there were 64 different proteins identified in COVID-19 patients across all severities compared to COVID-19-negative controls (Figure 2E). Most of these proteins were downregulated by COVID-19, especially in the severe group (Figure 2E). The alpha-2-HS-glycoprotein (*AHSG*) and the hepatocyte growth factor activator (*HGFAC*) were the only two proteins downregulated by COVID-19 across all severity groups (Figure 2E); the rest of the proteins were differentially expressed in a severity-dependent manner. The lists of differentially expressed proteins per severity group, including their accession ID, fold-change, and *p*-value, are shown in Appendix A.

The list of differentially expressed proteins in mild, moderate, and severe COVID-19 positive compared with COVID-19-negative patients are included in Appendix A, respectively. For mild COVID-19 patients, the six upregulated proteins were related to the oxygen carrier hemoglobin and acute phase immune responses, while three downregulated proteins were related to stress responses and metabolism. For moderate COVID-19 patients, the three upregulated proteins were hemoglobin, while the eight downregulated proteins were related to cell organization, stress responses, and metabolism. For severe COVID-19 patients, the only two upregulated proteins were acute phase proteins and hemoglobin, like mild COVID-19-positive patients, while the 54 significant downregulated proteins included proteins involved in cell adhesion, stress responses, and metabolic processes.

After performing a canonical pathway enrichment analysis on proteins associated with severe COVID-19, we found that the top 10 significantly enriched pathways were the following: LXR/RXR activation, FXR/RXR activation, acute phase response signaling, coagulation system, intrinsic prothrombin activation pathway, iron homeostasis signaling pathway, atherosclerosis signaling, neuroprotective role of THOP1 in Alzheimer’s Disease, MSP-RON signaling in cancer cells pathway, and leukocyte extravasation signaling (Figure 3).

In the activity prediction analyses, results showed that for the Mild and Moderate COVID-19 groups, neither a positive nor negative z-score was found for the significantly enriched canonical pathways (Figure 4A,B). For the severe COVID-19 group, based on their negative z-scores, the following pathways are predicted to be inhibited ordered by the magnitude of the z-core (higher blue intensity): LXR/RXR activation (z-score: −2.714), production of nitric oxide and reactive oxygen species in macrophages (z-score: −2.000), synaptogenesis signaling pathway (z-score: −2.000), and coagulation system (z-score: −0.447) (Figure 4C).

### 2.3. Validation of Cadherin-13 as Potential Biomarker of Severe COVID-19 in Puerto Ricans

For validation, we established our principal selection parameters in the following order: (1) Top three canonical pathways ordered by the higher absolute value of z-score in severe disease (Figure 4C; LXR/RXR activation: z-score: −2.714, production of nitric oxide and reactive oxygen species in macrophages: z-score: −2.000, and synaptogenesis signaling pathway: z-score: z-score: −2.000). (2) Literature search of the top three dysregulated proteins ordered from higher to lower fold change value, which participate in each of the top three canonical pathways and their association with severe COVID-19 (Appendix A). (3) From this group, we selected proteins not previously reported to be associated with severe COVID-19 in other cohorts (Appendix A). Using these parameters, we identified cadherin-13 as a candidate for validation using ELISA (Appendix A). Results confirmed that cadherin-13 was significantly decreased (*p* < 0.05) in the plasma of severe COVID-19 patients compared to healthy controls in our cohort (Figure 5A). As an additional validation parameter, we randomly selected other significant proteins from the list of 56 deregulated proteins in severe disease, which included the following: *PON1*, *KNG1*, hemoglobin, SAPC, *APOA2*, ICAM-1, and L-selectin (Figure 5). We ended up with eight proteins selected for validation using ELISA (Appendix A). There was a significant decrease in *PON1* levels in Moderate patients compared to healthy controls (*p* ≤ 0.05) (Figure 5B). There was a tendency towards a decrease in plasma PON1 levels in severe patients compared to healthy controls (*p* = 0.0575) and Mild patients (*p* = 0.0734). Similarly, for *KNG1*, there was a significant decrease in moderate patients compared to healthy controls (*p* < 0.01). There was a tendency towards a significant decrease in *KNG1* levels in mild (*p* = 0.0979) and severe (*p* = 0.0971) patients compared to healthy controls (Figure 5C). In contrast, for hemoglobin, there was a tendency towards a significant increase in moderate patients compared to healthy controls (*p* = 0.0663; Figure 5D). For SAPC (Figure 5E), *APOA2* (Figure 5F), ICAM-1 (Figure 5G), and sL-selectin (Figure 5H), there were no significant differences among the groups.

### 2.4. Cytokine Profile of Puerto Rican COVID-19 Patients

COVID-19-positive patients presented a significant elevation of three (*n* = 3) of the following cytokines in plasma compared to negative controls: IL-1Ra, IP-10, and TNF-α (Figure 6A). On the other hand, a significant reduction in PDGFb was observed in these patients (Figure 6A). When stratified by severity, no significant differences were observed for PDGFb between the groups. However, a significant decrease in PDGFb was observed in mild patients compared to negative controls (Figure 6B). For TNF-α, we found that severe patients had decreased levels of this cytokine in plasma compared to mild patients, whereas an increase was observed for mild patients compared to negative controls (Figure 6B).

## 3. Discussion

In this study, we performed proteomics and cytokine analyses to identify host factors associated with COVID-19 severity in Puerto Rico. Using proteomics studies and further validation through ELISA, we identified cadherin-13 as a potential biomarker of severe COVID-19 disease in Puerto Ricans. To our knowledge, this is the first study to demonstrate an association of decreased cadherin-13 levels with COVID-19 disease severity.

Quantitative proteomics analyses identified 64 differentially expressed proteins between COVID-19-positive individuals compared to negative controls. These included two proteins upregulated in Severe COVID-19 cases, while the great majority (54/56) were downregulated compared to negative controls. Among the upregulated proteins, serum amyloid P component (SAPC or *APCS*) or related proteins (*SAA*) have been found in proteomics studies from countries like Italy [23,24], India [25], China [21], Germany [26], Spain [27], and Saudi Arabia [28]. SAPC is an acute response protein linked to amyloid plaque accumulation that contributes to neuropathology, one of the manifestations of long-COVID-19. However, SAPC was also found to be upregulated in mild COVID-19. Among the 54 downregulated proteins found in this study, *APOA* and *APOF* were common to other studies in the world that include apolipoproteins (*APOC1*, *APO2*, *APO3*, *APOD*, *APOM*) [20,26,27,28]. Lipid metabolism proteins have been reported to contribute to COVID-19 disease severity [29]. Additional decreased proteins found in this study that are common to other studies include peptidase inhibitor 16 [30]. This protein is protective for the heart, one of the affected organs in COVID-19 disease. Gelsolin, a modulator of inflammatory responses, was another protein found to be downregulated in this study that has been associated with disease severity in other studies [22,24,27,28,31,32,33]. These proteins were not dysregulated in mild or moderate COVID-19-infected patients in our study.

In terms of canonical pathways associated with disease severity in our population, our results showed that the top four deregulated mechanisms based on the *p*-value were the following: LXR/RXR signaling, FXR/RXR signaling, the acute phase response signaling, and the coagulation system. However, based on z-scores or predicted activity, our results showed that the following pathways were inhibited in severe patients: LXR/RXR signaling, production of nitric oxide and ROS from macrophages, synaptogenesis signaling, and the coagulation system. Inhibition of the LXR/RXR signaling was recently reported in serum from COVID-19 patients in a longitudinal multi-omics study [34]. Oxidative stress is a mechanism that has been associated with the pathogenesis and severity of COVID-19, and antioxidants have been proposed as potential therapy [35,36]. The synaptogenesis signaling pathway was found to be inhibited in a meta-transcriptomics pathway enrichment analysis from bronchoalveolar lavage fluid collected from COVID-19 patients in hospitals in Wuhan in January 2020 [37]. However, cadherin-13 was not identified as deregulated in their cohort. Other studies have demonstrated that SARS-CoV-2 entry into neurons impairs synaptogenesis [38].

For validation studies, we established the following criteria: (1) top three canonical pathways ordered by higher absolute value of z-score in severe disease, (2) literature search of top three deregulated proteins based on higher fold change that participate in each of the top three canonical pathways and their association with severe COVID-19; and (3) selection of proteins not previously reported to be associated with severe COVID-19 in other cohorts. Following these parameters, we identified cadherin-13, also known as T-cadherin or H-cadherin. Cadherin-13 is a receptor for LDL and adiponectin and participates as a guidance receptor for the regulation of axons and blood vessel growth [39]. This receptor affects lipid and insulin metabolism and insulin sensitivity and plays a role in disorders such as in dyslipidemia, diabetes, obesity, insulin resistance, and atherosclerosis [39]. Cadherin-13 is an important regulator of GABAergic synapses [40]. In vitro cadherin-13 participates in the formation of excitatory and inhibitory synapses of hippocampal neurons [41]. Decreased levels of cadherin-13 in plasma are associated with increased severity of coronary artery disease and a higher risk of acute coronary syndrome [42]. Our validation results showed that COVID-19 patients had lower plasma levels of cadherin-13 in a severity-dependent manner and reached statistical significance in the severe group. Our results are in line with a recent genome-wide association study performed in a multi-center of European COVID-19 patients, which identified nine single nucleotide polymorphisms (SNPs) in the *CDH13* locus on chromosome 16, associated with COVID-19 risk of death [43]. These SNPs were associated with lung function and repair [43]. However, they could not control their data for the patients’ comorbidities due to the unavailability of data and a relatively small number of participants. Our cohort is controlled for comorbidities, except for cardiovascular disease, which was present in a significantly greater proportion in the severe group (3 out of 8 patients). Due to our small number of severe COVID-19 patients, we could not control for this comorbidity either. However, none of these three patients with cardiovascular disease were included in the TMT Labeling proteomic studies (Appendix A), and cardiovascular disease did not impact cadherin-13 levels in plasma (Appendix A). Similarly, despite the tendency towards a significant difference in age between the groups in our cohort (Table 1 and Appendix A), cadherin-13 levels did not correlate with age (Appendix A). Therefore, the decreased levels of cadherin-13 found in the severe COVID-19 patients by proteomics and ELISA cannot be attributed to cardiovascular disease but to an increased severity of COVID-19. In addition, our results suggest that COVID-19 patients with cardiovascular disease in our population may be more likely to develop severe disease due to COVID-19-induced depletion of cadherin-13 levels. The decrease of cadherin-13 in COVID-19 patients has not been reported before, suggesting that it could be a Puerto Rican-specific immune response to SARS-CoV-2. However, future studies evaluating cadherin-13 in a higher number of Puerto Rican participants are warranted to confirm these results. Variants on the *CDH13* gene associated with COVID-19 risk of death were recently reported in a European population [43]. It remains to be tested if catherin-13 protein levels are also decreased in Europeans. As Puerto Ricans from Puerto Rico have a high proportion of European (Spanish) ancestry (72%), mixed with ~15% African and ~13% Native American [44], we could expect that future studies evaluating single-nucleotide polymorphisms (SNPs) in our cohort could reveal unique variants on the *CDH13* gene that may affect its expression in response to SARS-CoV-2.

As a secondary validation approach, we randomly selected proteins from the list of significantly deregulated proteins in severe COVID-19 vs. healthy controls. These were the following: *PON1*, *KNG1*, hemoglobin, SAPC, *APOA2*, ICAM-1, and L-selectin. *PON1* and *KNG1* levels significantly decreased in moderate patients compared to controls and showed a tendency towards a significant decrease in our severe COVID-19 patients compared to healthy controls. *PON1* findings are consistent with previous studies that have demonstrated that decreased *PON1* levels are associated with disease severity [26,33,45,46]. A tendency towards a significant increase in hemoglobin levels for the moderate group was observed. This finding is consistent with the elevation of hemoglobin in proteomics data at all severity groups. In addition, abnormal hemoglobin levels are associated with the severity and death associated with COVID-19 [47].

In our study, we found a dysregulation of cytokines such as IL-1RA, TNF-α, IP-10, and PDGFb, which are elevated in combination in the plasma of COVID-19 patients in other studies and correlate with disease severity [6,48]. However, in our cohort, we found decreased PDGFb, especially in mild patients compared to controls. This is consistent with a previous study that found decreased serum PDGFb levels in COVID-19 patients compared to controls [49]. A recent study found that lower levels of PDGFb in the serum of COVID-19 patients compared to healthy controls within the first 24 h of hospitalization predict mortality [50]. However, other studies have found increased plasma levels of PDGFb in COVID-19 patients [51]. As PDGFb participates in tissue repair and angiogenesis, our results may indicate that COVID-19 might be inducing vascular tissue damage in our patients. In our cohort, TNF-α was elevated in mild COVID-19 and decreased with severity. This finding is supported by other studies that found decreased TNF-α levels with severity [52,53]. However, other studies have found elevated TNF-α levels in severe COVID-19 [54]. Finally, non-significant differences between disease severities were observed for IP-10 and IL-1Ra. IP-10 has also been found to be elevated in severe COVID-19 in other studies [54,55]. Our findings support previous findings about the heterogeneity of host factors associated with COVID-19 severity in different populations. This study was limited by a small number of patients recruited during the initial COVID-19 pandemic. Larger studies, including vaccinated participants and comparisons with other populations, are warranted to identify and validate risk factors for severity that are specific to our Puerto Rican population. In conclusion, although this cohort that started in 2021 as a pilot project was relatively small, the combination of proteomics and cytokine studies revealed important things about the factors that influence disease severity in our population. This study does not consider the role of viral strains as it was not its purpose but is an important determinant in the host response to the virus. The patients were recruited before Omicron variants emerged, and therefore, additional studies could reveal additional differences.

## 4. Materials and Methods

### 4.1. Study Participants, Ethics, and Sample Collection

This study includes individuals (*n* = 95), thirty-nine unvaccinated COVID-19 positive (*n* = 39), and fifty-six negative controls (*n* = 56) (Figure 1). Informed consent was obtained from each study subject before sample collection (IRB Protocol #0720120). Exclusion criteria were applied to subjects less than 21 years of age and participants who were unwilling or incapable of giving informed consent. Each patient completed an abbreviated form of the Columbia COVID-19 Questionnaire that was translated to Spanish–Patient Case Proband Version 2.3 (https://www.columbiamedicine.org/div/sons/kiryluk/COVID19/Columbia_COVID19_questionnaire_V2.3_ENGLISH.pdf, accessed on 1 September 2020), to collect clinical information, assess history of COVID-19 disease (symptoms, disease progression and treatment), risk factors (such as comorbid conditions), and outcomes. Subjects were stratified into four categories: negative controls and three observation conditions dependent on COVID-19 disease severity. The observation conditions were mild, moderate, and severe. Mild cases were defined as the individuals that tested positive for SARS-CoV-2 by PCR but had none or less than 6 symptoms associated with COVID-19 (loss of taste and smell, sore throat, fever, cough, malaise, headache, muscle pain, nausea, vomiting, diarrhea, shortness of breath). Moderate cases were defined as patients who had 6 to 12 symptoms of COVID-19 or those requiring an oxygen mask without the need for hospitalization. Severe cases were defined as adults who had 6 or more symptoms and were hospitalized, requiring invasive mechanical ventilation. This criterion is in accordance with CDC criteria, which defines severe outcomes of COVID-19 as hospitalization, admission to the intensive care unit (ICU), intubation or mechanical ventilation, or death [56]. As mild to moderate cases did not require hospitalization, we decided to differentiate these two groups based on the number of symptoms as described above. Those with mild and moderate disease were accrued from the University of Puerto Rico Laboratory of Parasite Immunology and Pathology (LPIP) after COVID-19 testing. Patients with severe disease were accrued from Auxilio Mutuo Hospital in collaboration with the Infectious Disease Physician, Dr. Jorge Bertran. Samples were transported to LPIP for processing and storage of saliva, plasma, and PBMC obtained from two 10 mL EDTA-blood tubes. Cells and saliva were saved for future genomics studies, and the plasma fractions were used for cytokines, chemokines, and proteomics studies. Study data were collected and managed using REDCap^®^ electronic data capture tools hosted at the University of Puerto Rico, Medical Sciences Campus [57,58].

### 4.2. Proteomics

#### 4.2.1. Depletion of Most Abundant Proteins

Plasma aliquoted samples from 22 (*n* = 22) unvaccinated COVID-19 patients stratified by disease severity and negative controls were randomly selected for quantitative proteomics studies (Appendix A). These included: COVID-19 patients with mild (*n* = 8), moderate (*n* = 10), and severe (*n* = 4) disease compared to eight (*n* = 8) COVID-19-negative controls. Briefly, the most abundant proteins, i.e., albumin and Immunoglobulin G (IgG), were removed from the samples using the Pierce Albumin/IgG removal kit (Thermo Fisher Scientific, Mount Prospect, IL, USA). Protein concentrations were determined using the bicinchoninic acid assay (BCA) (DC Protein Assay, Bio-Rad, Hercules, CA, USA), and 100 µg were aliquoted for the next step. Samples were then subjected to acetone precipitation overnight at −20 °C, centrifuged at 10,000× *g* for 10 min, and the supernatants discarded. Protein pellets were resuspended in 95% Laemli with 5% β-mercaptoethanol sample buffer and loaded into Mini-PROTEAN TGX precast gels. The gels were run for 15 min at 150 V and Coomassie Blue-stained. Gels’ images are shown In Appendix A. Thereafter, the gel lanes were diced manually into 1 mm3 cubes. The cubes were then de-stained using a solution of acetonitrile (50%) and ammonium bicarbonate (50 mM). Then, the samples were alkylated with iodoacetamide (10 mM) in ammonium bicarbonate (50 mM) and reduced using dithiothreitol (25 mM) in ammonium bicarbonate (50 mM). Samples were digested at 37 °C overnight with Trypsin, and peptides were gel-extracted using 150 μL of a mixture of acetonitrile (50%) and formic acid (2.5%) in water followed by 150 μL of acetonitrile (100%).

#### 4.2.2. TMT Labeling, Fractionation, and Mass Spectrometry Analyses

TMT labeling, fractionation, and mass spectrometry analyses were performed according to our previous studies [59,60]. Dried peptides were reconstituted in triethylammonium bicarbonate (100 mM) buffer and subsequently labeled with the TMT11-plex, followed by one hour of incubation. After 15 min of a quenching step, equal amounts of each sample per kit were mixed to generate a final pool that was later submitted to fractionation. The fractionation was performed using the Pierce High pH Reversed-Phase Peptide Fractionation Kit (Thermo Fisher Scientific, Mount Prospect, IL, USA) and following manufacturer’s instructions. Briefly, the column was conditioned twice using 300 μL of acetonitrile, centrifuged at 5000× *g* for 2 min, and the steps were repeated using 0.1% Trifluoroacetic acid (TFA). Each TMT labeled pool was reconstituted in 300 μL of 0.1% TFA and loaded onto the column. The bounded sample was washed and then eluted 8 times into 8 different vials using a series of elution solutions with different acetonitrile/0.1% triethylamine percentages indicated by the manufacturer. Then, each fraction was submitted to mass spectrometric analysis. Peptide separation was performed using high-performance liquid chromatography (Easy nLC 1200) (Thermo Fisher Scientific, Mount Prospect, IL, USA). Peptides were loaded onto a Pico Chip H354 REPROSIL-Pur C18-AQ 3 μM 120 A (75 μm × 105 mm) chromatographic column. The separation was obtained using a gradient of 7–25% of 0.1% of formic acid in acetonitrile (Buffer B) for 102 min, 25–60% of Buffer B for 20 min, and 60–95% Buffer B for 6 min. Making a total gradient time of 128 min at a flow rate of 300 nL/min. Separated peptides were analyzed using a Q-Exactive Plus mass spectrometer (Thermo Fisher Scientific, Mount Prospect, IL, USA). The instrument was operated in positive polarity mode and data-dependent mode. The MS1 (full scan) was measured over the range of 375 to 1400 *m*/*z* and at a resolution of 70,000. The MS2 (MS/MS) analysis was configured to select the ten most intense ions for HCD fragmentation at a resolution of 35,000. A dynamic exclusion parameter was set for 30.0 s.

#### 4.2.3. Protein Identification and Quantitation

Protein identification and quantitation were performed according to our previous studies [59,60]. Proteome Discoverer version 2.5 (Thermo Fisher Scientific, Mount Prospect, IL, USA) was used to identify and quantify proteins. Proteins were identified using a Human protein database obtained from the software protein center tool using tax ID = 9606. Protein identifications were obtained using a SEQUEST HT algorithm. The modifications included were the following: a dynamic modification for oxidation +15.995 Da (M), a static modification of +57.021 Da (C), and static modifications from the TMT reagents +229.163 Da (Any N Term, K). The false discovery rate was set at 0.01 (strict) and 0.05 (relaxed). A differential expression analysis was conducted using the R-Limma package (version 3.41.15) on Bioconductor version 3.16 [61]. COVID-19 severity groups (mild, moderate, and severe) were considered the observation groups and compared to the COVID-19-negative control group. In this study, the proteins considered significant were those with a fold change ≥ |1.5| and a *p*-value ≤ 0.05. Canonical pathway analyses were conducted for the significantly differentially expressed proteins for each observation condition using Ingenuity Pathway Analysis (IPA, version 22.0.2, QIAGEN Digital Insights, Germantown, MD, USA).

### 4.3. Cytokines

Plasma aliquots from 39 COVID-19-positive individuals stratified by disease severity and 56 COVID-19-negative subjects were obtained from the prospectively collected samples stored at −80 °C in the LPIP for determination of cytokines. Samples were matched for age and gender. Approximately 50–200 μL per sample were used to identify and quantify 23 cytokines, chemokines, and growth factors in plasma, using the Bio-Plex Pro Human Cytokine Screening Panel, following manufacturer instructions (Bio-Rad, Hercules, CA, USA). The following analytes were measured: IL-1β, IL-1ra, IL-4, IL-6, IL-7, IL-8, IL-9, IL-10, IL-12p70, IL-13, IL-17A, Eotaxin, FGF basic, G-CSF, GM-CSF, IFN-γ, IP-10, MCP-1, MIP-1α, PDGF-BB (or PDGFb), MIP-1β, RANTES, and TNF-α. Values below the lower limit of detection, not detected, and/or outliers (ROUT, Q = 1%) were excluded from further analyses.

### 4.4. Validation Using ELISA

Repository plasma samples from unvaccinated healthy controls (*n* = 8) and COVID-19-positive patients stratified by disease severity (Mild: *n* = 12, Moderate: *n* = 12, and Severe: *n* = 8) were randomly selected for validation using ELISA, following manufacturer instructions. Random selection of proteins for validation is an accepted approach in TMT-labeling proteomics studies [62,63,64]. The following ELISA analyses were performed: cadherin-13 (Invitrogen, Waltham, MA, USA), *PON1* (Ray Biotech, Peachtree Corners, GA, USA), *KNG1* (Ray Biotech, Peachtree Corners, GA, USA), hemoglobin (Sigma Aldrich, St. Louis, MO, USA), SAPC (Abcam, Waltham, MA, USA), *APOA2* (Abcam, Waltham, MA, USA), ICAM-1 (Abcam, Waltham, MA, USA), and sL-selectin (Invitrogen, Waltham, MA, USA).

### 4.5. Statistical Analyses

For analyses of participants’ demographics, Kruskal–Wallis test was performed to determine differences in age between the groups, and Fisher’s exact test was performed to determine differences in categorical variables such as sex, Hispanic origin, and presence of comorbidities. For TMT labeling proteomics bioinformatic analyses, the statistical analysis was performed pairwise between the groups, as follows: Case vs. Controls. Differentially expressed proteins were considered significant with a fold change (FC) ≥ |1.5| (Log2 FC ≥ |0.5|) and *p*-value < 0.05. Identified proteins for each comparison were used to generate the heatmap. For the volcano plot, the threshold was set as –log10 *p*-Value ≥ 1.30 (or *p*-value ≤ 0.05) and FC ≥|1.5| (Log2 FC ≥ |0.5|). For cytokine and protein validation analyses, outliers (ROUT, Q = 1%) were eliminated from raw data. Normal distribution of data was assessed using Shapiro–Wilk, and parametric or non-parametric tests were performed where appropriate. Unpaired *t*-tests were performed for comparisons regarding COVID-19 status (positive vs. negative) or for comparisons of each severity group vs. the healthy control group. For comparisons between the different disease severities (Mild, Moderate, and Severe), One-way ANOVA or Kruskal–Wallis tests were performed where appropriate. Statistical significance was considered at *p* < 0.0500. A schematic of our study design is shown in Figure 1.

## Figures and Tables

**Figure 1 ijms-25-05426-f001:**
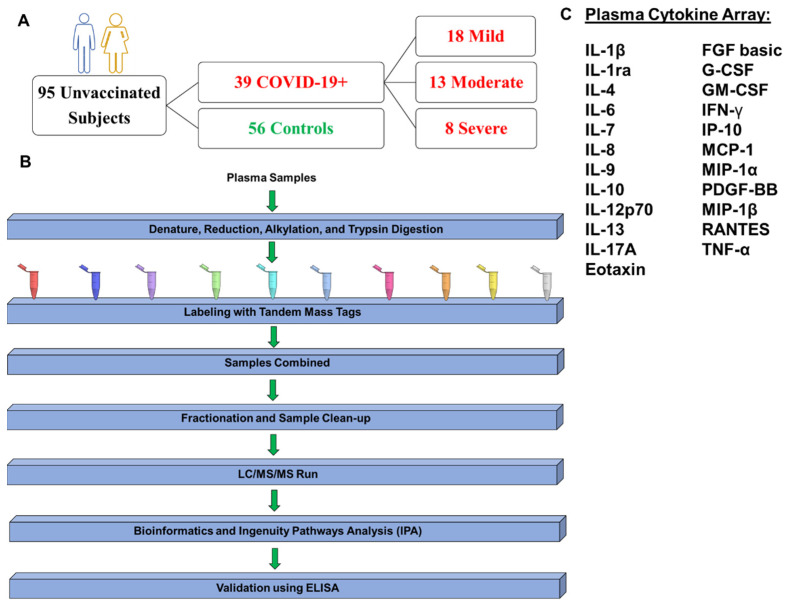
Study design. This protocol was approved by the UPR-MSC IRB (#0720120) and Biosafety Committees. (**A**) Ninety-five (95) unvaccinated men and women between the ages of 21 and 71 were recruited in Puerto Rico. Plasma samples were collected from thirty-nine unvaccinated COVID-19 positive (*n* = 39) and fifty-six negative controls (*n* = 56). COVID-19 patients were stratified based on symptomatology as follows: mild (*n* = 18), moderate (*n* = 13), and severe (*n* = 8). (**B**) Quantitative proteomics studies were performed in plasma samples from thirty (*n* = 30) participants using tandem mass tag (TMT). These included twenty-two (22) randomly selected COVID-19-positive patients with mild (*n* = 8), moderate (*n* = 10), and severe COVID-19 disease (*n* = 4) that were matched to COVID-19-negative controls (*n* = 8). Proteins were isolated, denatured, reduced, alkylated, digested, and labeled using TMT at the Translational Proteomics Center. Labeled peptides were subjected to mass spectrometry and analyzed by Proteome Discoverer (version 2.5), *Limma* software (version 3.41.15), and Ingenuity Pathways Analysis (IPA, version 22.0.2). Most relevant proteins associated with COVID-19 severity were interrogated by ELISA. (**C**) Twenty-three pro-inflammatory cytokines, chemokines, and growth factors associated with a severe COVID-19 outcome were identified and quantified from all subjects (*n* = 95) using a human cytokine array.

**Figure 2 ijms-25-05426-f002:**
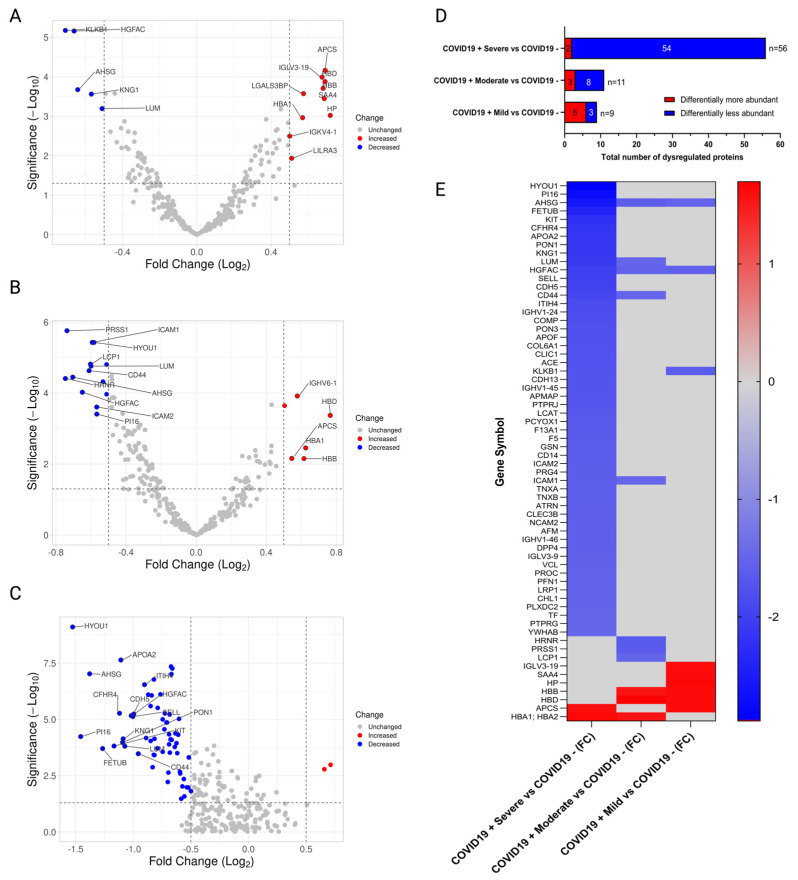
Differentially expressed proteins according to COVID-19 severity. (**A**) Volcano plot comparing between Mild COVID-19 positive (+) vs. COVID-19 negative (−) controls; (**B**) Volcano plot for comparison between Moderate COVID-19 positive (+) vs. COVID-19 negative (−) controls. (**C**) Volcano plot for comparison between Severe COVID-19 positive (+) vs. COVID-19 negative (−). All Volcano plots depict differentially abundant proteins identified per group comparison. (**D**) Stacked bar plot depicting dysregulated proteins (differentially more abundant or differentially less abundant) per each group comparison using a one-factor analysis between cases vs. controls (COVID-19 negative). (**E**) Heatmap showing FC of dysregulated proteins with a significance of *p*-value < 0.05. Blue color indicates downregulated proteins. Red color indicates upregulated proteins. Grey color indicates proteins that were not dysregulated in that comparison. All dysregulated proteins depicted in this panel complied with a threshold of (FC) ≥ |1.5| (Log2 FC |0.5|) and *p*-value ≤ 0.05 (−log10 *p*-Value ≥ 1.30) and are represented by their gene symbol.

**Figure 3 ijms-25-05426-f003:**
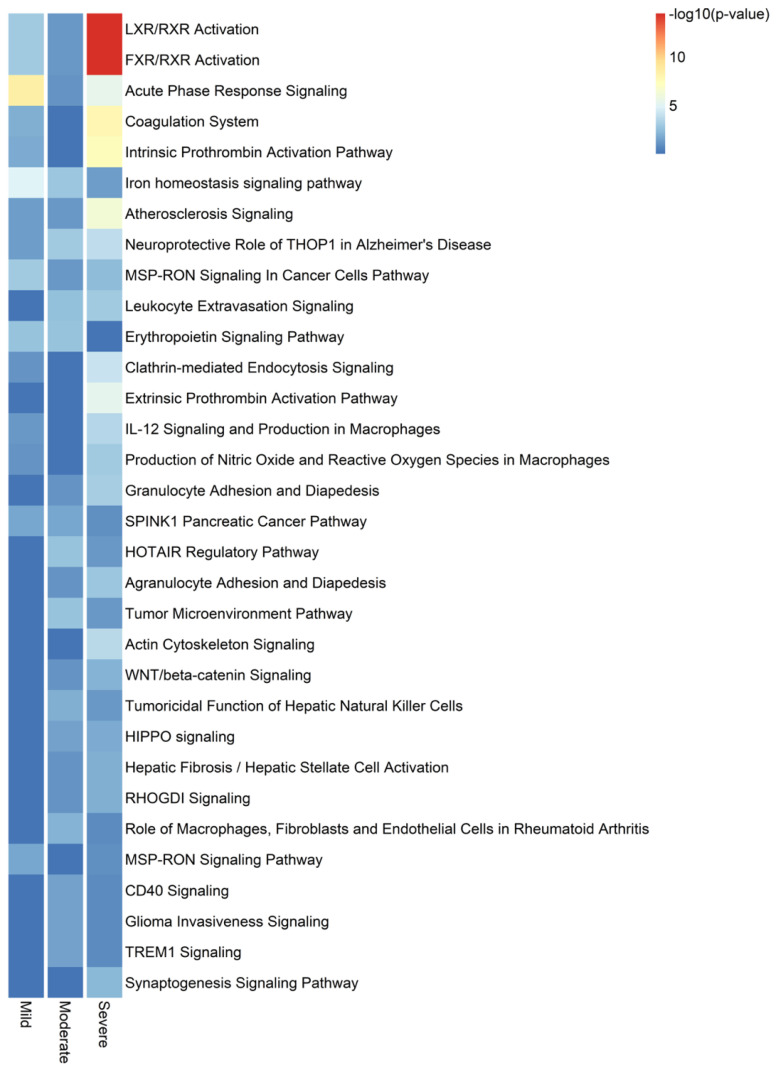
Top 32 canonical pathways differentially deregulated by COVID-19 disease severity. Heat map for enriched canonical pathways with a gradient-based IPA −log_10_ (*p*-value) for the significantly different proteins found in mild, moderate, and severe COVID-19 positive compared with COVID-19 negative groups.

**Figure 4 ijms-25-05426-f004:**
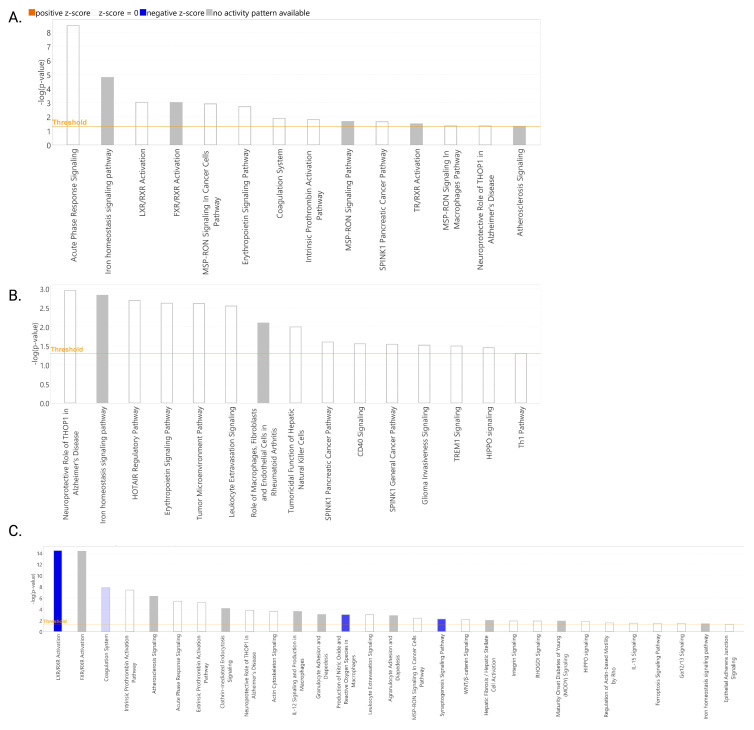
Predicted activity of canonical pathways in COVID-19 patients. Z-scores were computed by IPA for significantly enriched canonical pathways for Mild (**A**), Moderate (**B**), and Severe (**C**) COVID-19 groups. Blue bars indicate predicted inhibited pathways. Gray bars indicate pathways for which an activity prediction could not be made. White bars indicate pathways with a z-score close to or equal to 0. The intensity of the bar color correlates with the z-score prediction value. Pathways with a significant enrichment −log10 (*p*-value) ≥ 1.5 (*p*-value = 0.05) are shown. This image was obtained from IPA.

**Figure 5 ijms-25-05426-f005:**
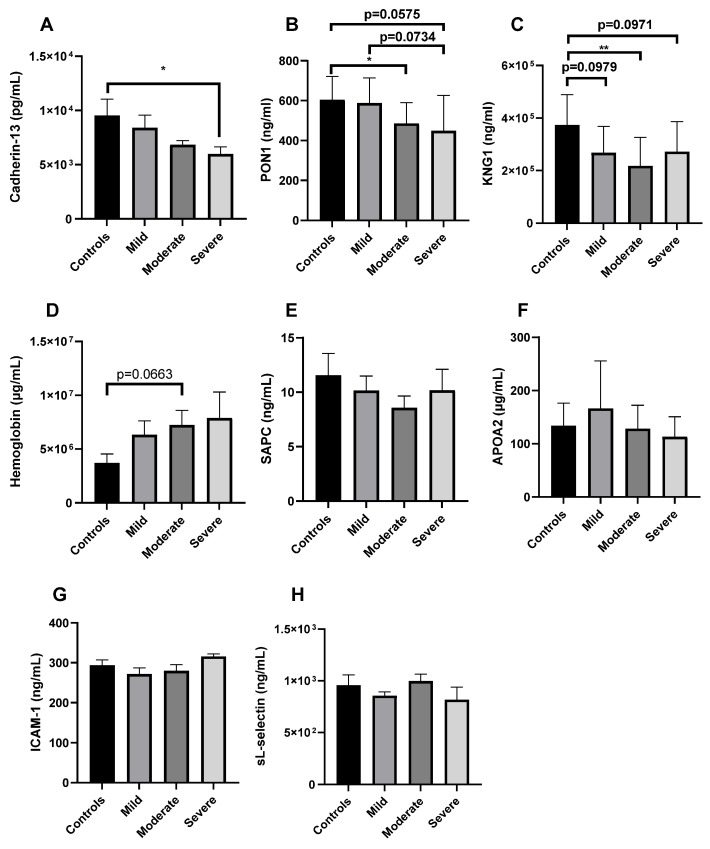
Validation of relevant proteins associated with COVID-19 disease severity. Plasma samples from unvaccinated healthy controls (*n* = 8) and COVID-19-positive patients stratified by severity (Mild: *n* = 12, Moderate: *n* = 12, and Severe: *n* = 8) were randomly selected for validation using ELISA of the following 8 proteins: cadherin-13 (**A**), *PON1* (**B**), *KNG1* (**C**), hemoglobin (**D**), SAPC (**E**), *APOA2* (**F**), ICAM-1 (**G**), and sL-selectin (**H**). The mean ± SEM is shown. * *p* < 0.05, ** *p* < 0.01.

**Figure 6 ijms-25-05426-f006:**
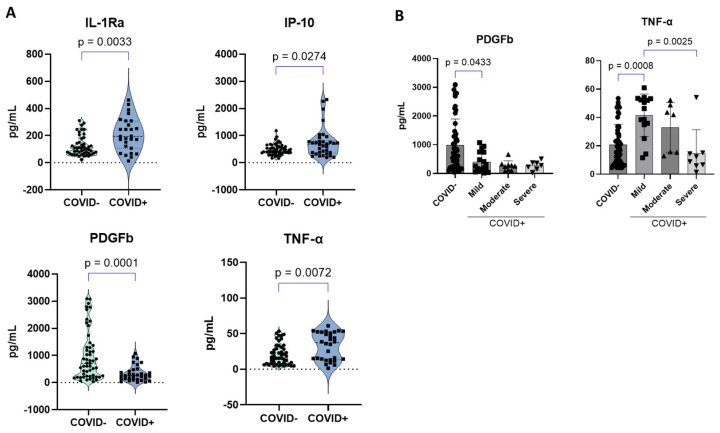
Plasma cytokines in study participants. Twenty-three plasma cytokines, chemokines, and growth factors were measured and quantified using a human cytokine array from plasma samples of participants (*n* = 95) using the Bio-Plex Pro Human Cytokine Screening Panel from (Bio-Rad, Hercules, CA, USA). (**A**) Significantly deregulated cytokines when stratified by COVID-19 status. (**B**) Significantly deregulated cytokines when stratified by COVID-19 severity. Any missing values or analytes were either considered outliers (ROUT, Q = 1%), not detected, and/or below the lower limit of detection. The mean ± SEM is shown.

**Table 1 ijms-25-05426-t001:** Demographics, comorbidities, and clinical symptoms of participants by COVID-19 severity. N/A = Does not apply.

	COVID-19 Negative Controls	COVID-19 (+) Mild	COVID-19 (+) Moderate	COVID-19 (+) Severe	Total (%)	*p*-Value
Number of participants	56	18	13	8	95	N/A
Mean age (range)	45 (21–71)	44 (28–56)	34 (23–53)	51 (28–67)	43 (21–71)	0.0584
Women	33	11	8	4	56 (58.9%)	0.9532
Men	23	7	5	4	39 (41.1%)	0.9532
Hispanics	51	18	13	8	90 (94.7%)	0.2987
Non-Hispanics	5	0	0	0	5 (2.6%)	0.2987
Autoimmune diseases	7	3	2	2	14 (14.7%)	0.8130
Cancer	2	1	0	0	3 (3.2%)	0.7881
Diabetes	4	2	0	1	7 (7.4%)	0.6333
Cardiovascular diseases	0	2	1	3	6 (6.3%)	0.0005
High blood pressure	8	5	1	2	16 (16.8%)	0.3985
HIV/AIDS	1	1	1	0	3 (3.2%)	0.6110
Lung disease	1	0	1	0	2 (2.1%)	0.4653
Kidney disease	1	2	1	0	4 (4.2%)	0.2997
Loss of smell	0	6	10	4	20 (21.1%)	N/A
Loss of taste	0	5	9	4	18 (18.9%)	N/A
Muscle aches	0	9	11	5	25 (26.3%)	N/A
Cough	0	14	8	6	28 (29.5%)	N/A
Shortness of breath	0	2	4	5	15 (30%)	N/A
Fever	0	6	9	6	21 (22.1%)	N/A
Headache	0	12	9	6	27 (28.4%)	N/A
Chest pain	0	0	4	3	7 (7.4%)	N/A

## Data Availability

Most of the data generated or analyzed during this study are included in this published article. All the proteomics raw datasets generated in the current study have been deposited in the ProteomeXchange [82] Consortium via the PRIDE [83], a partner repository with the following dataset identifier: Project Name: Protein and cytokine profiles associated with COVID-19 disease severity in Puerto Rico Project accession: PXD046103. Project DOI: 10.6019/PXD046103.

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
