# Peer review of "Plasma Proteins Associated with COVID-19 Severity in Puerto Rico"

_ijms, 2024, doi:10.3390/ijms25105426_

Round 1

Reviewer 1 Report

Comments and Suggestions for Authors

Comments and Suggestions:

The article “Plasma proteins associated with COVID-19 severity in Puerto Rico” by Rosario-Rodríguez et. al., provides an insight into the quantitative proteomics and cytokine analysis of plasma samples isolated from COVID-19-positive and -negative individuals to find host factors related to COVID-19 severity. They also showed that Cadherin-13 and TNF-α were downregulated in COVID-19 patients as confirmed by ELISA and cytokine immunoassay respectively.

Clarity and Structure: The introduction effectively sets the stage by presenting thorough explanation about COVID-19 history, immune responses and studies that identified many cytokines and proteins altered during disease severity. Authors have taken proper controls.

Language, Grammar and Figures: Review the text for any grammatical errors and ensure that the language used is clear and concise. Please ensure to upload higher resolution images/figures to have a clear view to understand it.

I do have some minor comments:

1.      Introduction: The authors must include few recent references which describes about the proteomics analysis and identification of important deregulated proteins during severe COVID-19 infection.

2.      Figure 1C and section 2.4: The kit used is able to detect 36 human cytokines, chemokines, and acute phase proteins simultaneously. The authors have mentioned only 23 of 36 in Figure 1C. Were they not detected or below the lower limit of detection of the method used? The remaining 13 are also important ones involved in immune response to COVID-19.

3.      Page 9, line 202: What was the criteria of selecting 8 random proteins for additional validation as out of 8 only 4 showed significant differences among all groups in validation. The authors could have selected other proteins such as ACE protein which is an important protein deregulated in COVID-19 infection and extensively studied.

4.      Page 14, line 372: On what basis 0-5, 6-12 and above 12 symptoms are selected for mild, moderate and severe cases respectively. Are there any references which confirms these criteria, please add it? Are there any specific symptoms that distinguish all the three cases?

5.      Section 4.2.2: The authors should cite proper reference.

6.      Page 1, line 24: Authors are suggested that, it is not necessary to mention the IRB Protocol approval Number in Abstract section. This is already mentioned in the manuscript.

7.      Figure 2A, 2B, 2C: It is difficult to understand the figures. Please upload it in higher resolution.

Author Response

See uploaded file

Reviewer 2 Report

Comments and Suggestions for Authors

Rosario-Rodríguez et al studied the proteomics profiling in COVID-19 patients in Puerto Rican patients. They hypothesized that the host response to SARS-CoV-2 could be ethnic population dependent, such as Puerto Ricans, which is a highly admixed population. Although in this study patients are mainly from Hispanic origin. They identified cadherin 13 and validated by ELISA that decreases with COVID-19 severity. This is a comprehensive study although the patient number is small for an ethnic specific conclusion. The differential protein profile is most likely the host response to SARs-CoV-2.  However, authors should modify/correct these following concerns before acceptance for publication.

1.      In abstract they studied 39 COVID-19 positive patients. But in line 444, authors stated they used 50 COVID-19 patients for cytokine analysis. From where these extra patients came, and they did not mention anywhere about these extra number of patients. They should correct this and modify the text appropriately wherever they mentioned the number of patients. Whether they are informed consented and IRB compliance?

2.      Figure 1C is not described in the text or not descriptively in the legend. So, it is difficult to understand whether these cytokines are increased or decreased in COVID-19 patients.

3.      The resolution of figures 2 and 4 is very low and illegible. They should considerably increase the resolution.

4.      Figure 2e, they should justify each color code for up or down regulation.

5.      Do the author think that decrease of cadherin 13 is a Puerto Rican specific host response to SARS-CoV2? As Peurto Rican had 72% European ancestry, did Italy, Spanish and German studies also detected CDH13 in their proteomic studies?  

Author Response

See file uploaded
